# *F5* 6665A>G Polymorphism Is Associated with Increased Risk of Venous Thromboembolism in Females

**DOI:** 10.3390/ijms26062403

**Published:** 2025-03-07

**Authors:** Sladjana Teofilov, Olivera Miljanović, Jelena Vuckovic-Filipovic, Natasa Djordjevic

**Affiliations:** 1Center for Medical Genetics and Immunology, Clinical Centre, 81 000 Podgorica, Montenegro; sladjana.teofilov@kccg.me (S.T.); olivera.miljanovic@kccg.me (O.M.); 2Department of Internal Medicine, Faculty of Medical Sciences, University of Kragujevac, 34 000 Kragujevac, Serbia; 3Department of Pharmacology and Toxicology, Faculty of Medical Sciences, University of Kragujevac, 34 000 Kragujevac, Serbia; natasa.djordjevic@fmn.kg.ac.rs

**Keywords:** venous thromboembolism, *F5* 6665A>G, females

## Abstract

The main goal of our study was to assess the potential effect of the polymorphism of the coagulation-related genes *F2*, *F5*, and *F13A* on the risk of venous thromboembolism (VTE) development. The study was conducted at the Clinical Center, Podgorica, Montenegro, and included 103 VTE patients and 106 sex- and age-matched healthy controls. Demographic, clinical, and laboratory data were obtained from the medical records and questionnaires. Genotyping for *F2* 19911A>G (rs3136516), *F5* 6665A>G (rs6027), and *F13A* 102G>T (rs5985) was performed by allele-specific PCR. Controlling for the effect of known risk factors, the presence of at least one variant *F5* 6665 G allele conferred a significantly higher risk of VTE among females [OR (95%CI): 64.06 (5.38; 763.61)], but not among males. In addition, thromboembolic events were associated with comorbidities [OR (95%CI): 197.10 (19.17; 2026.19)], overweight [OR (95%CI): 33.59 (2.47; 456.65)], and the presence of *F2* 20210G>A [OR (95%CI): 32.43 (4.21; 249.77)] and *F5* 1601G>A [OR (95%CI): 144.80 (13.59; 1542.63)] in females, as well as with comorbidities [OR (95%CI): 6.32 (1.90; 20.98)], family history of VTE [OR (95%CI): 8.10 (2.28; 28.83)], and the presence of *F5* 1601G>A [OR (95%CI): 20.10 (2.34; 173.02)] in males. Our study reports an association between the presence of at least one *F5* 6665G variant allele and an increased risk of VTE development in females. Our results indicate that *F5* 6665A>G, in combination with other confirmed factors of influence, such as comorbidities, overweight, *F2* 20210G>A, and *F5* 1601G>A, could contribute to VTE risk prediction in females.

## 1. Introduction

Venous thromboembolism (VTE) is a pathological condition characterized by hypercoagulability and the accompanying formation of thrombi in the veins, which impairs venous circulation and leads to obstruction [1]. This multifactorial disorder arises from a complex interaction of numerous acquired and inherited risk factors [1,2,3,4], with genetic predisposition being held responsible for up to 60% of all thromboembolic events [1,5,6,7,8].

Genes implicated in VTE development usually code for components of the coagulation/anticoagulation pathways [9]. Among them, *F2* and *F5* [5], which are responsible for the synthesis of prothrombin (Factor II, FII) and proaccelerin (Factor V, FV), respectively [10,11], stand out as the most frequently investigated in this regard [8]. Prothrombin is a precursor of the serine protease thrombin, which has a central role in the coagulation process: it catalyzes the transformation process of fibrinogen into fibrin monomers, increases the permeability of the endothelium, and stimulates the aggregation of platelets [12,13]. Its coding gene *F2* features specific variation in the 3′ untranslated region, 20210G>A, which was, from its first description in 1996 [14], almost invariably associated with the risk of thromboembolism [9]. Possible roles of other *F2* polymorphisms, including frequent and functional intronic variant 19911A>G, have been assessed, but the results of the studies were inconsistent [15,16,17]. On the other hand, proaccelerin has a dual role in the process of hemostasis, which depends on activation and interaction with other coagulation factors: in a form of precursor, it acts as an anticoagulant [18]; once activated, it acts as a cofactor for the prothrombinase complex that converts prothrombin to thrombin [19,20,21,22]; when it succumbs to further alterations, it regains anticoagulant properties [23,24,25]. The *F5* gene that codes for proaccelerin is highly polymorphic, and its variations are usually functional [26,27]. The most important *F5* polymorphism associated with an increased risk of VTE is 1601G>A (legacy numbering 1691G>A [28]), known as the “Leiden mutation” [21,29,30,31,32]; it was discovered in 1994 [30] and ever since has been considered the most common defect in patients with thromboembolic disease, caused by reduced FV degradation and the accompanying hypercoagulability [8,11,20,33]. Other *F5* variations have been examined too, and one of the most intriguing seems to be 6665A>G, located within the functional domain of the gene [19]; its role in VTE development is still unresolved [34,35]. The possibility of association between many other coagulation-related genes and susceptibility to VTE has been investigated as well. Available research suggests several genes to be of importance in this regard, including *F13A*, which codes for a functional component of the fibrin stabilizing factor (Factor XIII, FXIII). This factor has many functions in the coagulation cascade, but its crucial role, which depends upon its interaction with thrombin, is in the enhancing and stabilization of the cross-linking of fibrin into a clot [36]. *F13A* exhibits more than 100 different variations, usually associated with lessening or complete loss of fibrin stabilizing factor activity [37]. Among them, the most frequently studied, 102G>T, located near the thrombin activation site, has been reported as protective against venous thrombosis, but contradictory findings have been published as well [38,39,40].

Given the severity of this disease and the estimated significance of genetics in its development, as well as the unexpected inconsistencies in the literature regarding the most likely genetic factors, the main goal of our study was to assess the potential effect of *F2* 19911A>G, *F5* 6665A>G, and *F13A* 102G>T on the risk of VTE development, while controlling for the well-known role of the *F2* 20210G>A and *F5* 1601G>A variants and the impact of the most important demographics and clinical characteristics.

## 2. Results

The characteristics of the study participants, as well as the results of the univariate logistic regression analysis assessing the risk of VTE development associated with each of the available variables, are presented in Table 1. A comparison between the patients and controls revealed that BMI, overweight (defined as BMI above 25), comorbidities (arterial hypertension, coronary disease, diabetes mellitus, surgical interventions, or trauma), and family history of VTE increased the risk of disease. Other variables, including sex, age, cigarette smoking (classified as ever or never), coffee or alcohol consumption, and daily physical activity (based on both work and leisure), did not demonstrate a significant effect on VTE development.

Minor allele frequencies, genotypes, and genotype groups (dominant and recessive genetic models) among the study participants are presented in Table 2. All the genotypes were in Hardy–Weinberg equilibrium (HWE) (χ2 (1) < 3.288, *p* > 0.193). Univariate logistic regression analysis, assessing the effect of investigated SNPs, genotypes, and genotype groups on the risk of VTE development (Table 2), did not reveal any significant association. 

Subsequent multivariate logistic regression with backward elimination, starting from a full model combining all potentially significant variables (including genotype data for *F2* 20210G>A and *F5* 1601G>A), revealed a difference in the VTE risk factors between males and females (Table 3). The best-fitting regression model in females (Hosmer–Lemeshow: χ2(4) = 0.052, *p* = 1.000) was statistically significant (χ2(5) =95.513, *p* < 0.001); it explained 78.7% of the variance in the risk of VTE development and correctly classified 91.6% of cases. According to the model, comorbidities, overweight, and the presence of at least one variant *F2* 20210A, or *F5* 1601A or 6665G allele, were all associated with an increased likelihood of VTE development in females. On the other hand, the best-fitting regression model in males (Hosmer–Lemeshow: χ2(3) = 3.325, *p* = 0.344) that was statistically significant (χ2(3) = 42.790, *p* < 0.001) explained 45.7% of the variance in the VTE risk and correctly classified 81.4% of cases. According to this model, family history of VTE, comorbidities, and the presence of at least one variant 1601A allele of *F5* were the only factors, among those that were investigated, to affect the risk of VTE development in males.

## 3. Discussion

In the present study, we analyzed the potential association between VTE and the polymorphism of three different coagulation-related genes, namely *F2* 19911A>G, *F5* 6665A>G, and *F13A* 102G>T. Controlling for the effect of known acquired and inherited risk factors, our study showed that the presence of at least one variant *F5* 6665 G allele confers a significantly higher risk of VTE among females, but not among males. To the best of our knowledge, this is the first study reporting a sex-dependent connection between *F5* genetic polymorphism and VTE development. In addition, we confirmed the association of thromboembolic events with comorbidities, overweight, and the presence of *F2* 20210G>A and *F5* 1601G>A in females, as well as with comorbidities, family history of VTE, and the presence of *F5* 1601G>A in males.

The *F5* gene is located on chromosome 1q23; it spans a region of approximately 80kb and integrates 25 exons [18]. Its product, named proaccelerin or Factor V (FV), is a multidomain (NH_2_-A1-A2-B-A3-C1-C2-COOH) pre-procofactor of 2224 amino acids (aa), including 28 aa residue signal peptides [18,20,22]. FV is known for its significant and rather complex role in the coagulation cascade: as a part of the prothrombinase complex, it can enhance the prothrombin activation rate by about 300,000-fold [11,41]. However, to take part in this process, proaccelerin (initially an anticoagulant that regulates FVIIIa activity [18]) needs to be activated to accelerin (FVa), which can be achieved either by thrombin or by activated Factor X (FXa) [42]. It has been demonstrated that thrombin-dependent proteolytic cleavage, which eliminates the inhibitory B domain and activates FV, uses the C-terminal C2 domain as a scaffold [43,44].

In addition to its FV activation-related function, the C2 domain has also been proven to be essential for FVa procoagulant activity: it provides accelerin binding to the phosphatidylserine-exposing lipid membrane of activated platelets or endothelial cells, enabling formation of the prothrombinase complex with FXa [20,45,46,47]. At the same time, in its membrane-bound form, accelerin becomes a substrate for another protease known as activated protein C (APC), with which it can become further cleaved, regaining anticoagulant, i.e., FVIIIa-inactivating, properties [23,24,25,48]. Since the C2 domain represents a common denominator in all of the described processes, any *F5* variation located within or near its coding region could potentially affect both accelerin formation and its pro- or anticoagulant activities.

*F5* 6665A>G is a missense variant that leads to amino acid Asp>Gly replacement at position 2222 within the FV C2 domain, i.e., only three aa from the end of the molecule [28]. In our study, 6.5% of the participants were carriers of the variant 6665G allele, which fits well with the European frequency data provided by the 1000 Genomes Project [49,50] and published in the literature [35,51,52]. Based on the sequence homology and the differences between alternative amino acids, this change has been predicted to be deleterious, i.e., probably damaging to protein function (SIFT and PolyPhen scores of 0 and 0.98, respectively) [28,50], and the predictions were confirmed by in vitro experiments [53]. Specifically, *F5* 6665A>G neighbors the only disulfide bridge in the C2 domain [27,28], whose role is to secure the structure and the function of the domain by preventing it from unfolding [54]. It has been shown by Miteva et al. [55] that the presence of this variation destabilizes the C2 domain by inducing conformational changes and affecting the formation of the disulfide bond. Based on these observations, impaired FV secretion associated with the presence of Asp2222Gly substitution could be explained by protein misfolding, which results in its retention and subsequent degradation within the endoplasmic reticulum [53]. In addition, this variation is also located close to the accelerin glycosylation site at Asn2181 [18], which is responsible for differentiation between two main forms of FVa, namely FVa1 and FVa2: the former is more glycosylated and is thus associated with impaired membrane binding and lower APC-cofactor activity [18,20]. As the modulation of FVa glycosylation efficiency leads to an imbalance between its two forms, we assume that the increased risk of VTE in the presence of Asp2222Gly substitution could be explained by the dominance of FVa1 over its counterpart FVa2, displaying up to 7-fold higher thrombin generation and procoagulant action [56,57]. The results of our study conform well to the previous observations, with the risk of VTE development being higher in the presence of the *F5* 6665A>G variant allele.

However, the association of *F5* genetic variation and venous thromboembolism observed in our study was sex-dependent, as we were able to identify it only among females. The role of sex in VTE development has been frequently studied and described, with the incidence of the first episode prevailing either in females or in males, mostly depending on age [58,59]. The reasons for the observed deviation have been addressed profusely, and the myriad of potential explanations suggests that this phenomenon is most probably multifactorial [59]. In our study, the effect of sex on VTE risk was not detected, and we suspect this to be due to the small sample size and wide age range of our study groups. However, the separate analysis of male and female participants revealed a clear difference in several risk factors associated with VTE, including the presence of the *F5* 6665A>G variation.

The interdependence between genetic polymorphism and sex in affecting VTE development have been thoroughly investigated; however, none of the most important inherited risk factors, such as *F2* 20210G>A, *F5* 1601G>A, and deficiencies of antithrombin, protein C, and protein S [60,61], were declared important in sex-specific aspects of this disease [59]. This lack of association was explained by chromosomal location of genes: none of the examined genes were located on the sex chromosomes, thus they were neither expected nor found to be differentially distributed between females and males [58]. However, they all participate in the complicated pathway of the coagulation process, which also involves factors VIII (FVIII) and IX (FIX), whose coding genes (*F8* and *F9*, respectively) are located on the X chromosome [62,63]. In females, one X chromosome is normally inactivated, so X-linked gene expression is expected to be balanced between sexes [64]. Nevertheless, a certain limited number of genes, including *F8* [65], can escape X-chromosome inactivation (XCI) [66], thus contributing to sex-related difference in their activity, usually through female-biased gene expression. In line with its XCI status, FVIII displays higher levels in females compared to males [67].

The role of FVIII in the coagulation pathway is considered essential: in its activated form, it acts as a cofactor in the process of FX activation [62,68]. Due to its strong procoagulant activity, increased levels of FVIII, compared to other coagulation factors, were associated with the highest risk of venous thrombosis [69]. Proteolysis of FVIIIa, which results in its inactivation, mainly depends on APC anticoagulant activity, contributed by either an intact or inactivated form of FV [18,70]. Therefore, impaired proaccelerin secretion and lower accelerin APC-cofactor activity, both associated with the presence of *F5* 6665A>G variation [53,56,57], could lessen the efficacy of APC-dependent FVIII cleavage and extend its procoagulant capacity. The proposed explanation could clarify not only the observed association between *F5* 6665A>G and increased risk of VTE, but also its sex-specific aspects: in the case of impaired FVIIIa inactivation, higher levels of FVIII, consequent to female-biased *F8* gene expression, are expected to display significantly more thrombogenic phenotypes in females compared to males. Sexual dimorphism in disease prevalence due to sex-specific gene expression has already been reported, including the association between *CFTR* IVS8 poly-T variation and the severity of acute pancreatitis in women [71] and the sex-dependent role of *IFNL4* rs12979860 and rs368234815 polymorphisms in COVID-19-related pneumonia development [72]. The present study is the first to report an increased risk of VTE development in female carriers of the *F5* 6665A>G variation, indicating the potential predictive capacity of routine F5 genotyping in women.

In addition to *F5* 6665A>G polymorphism, in our study higher VTE risk was significantly associated with several other acquired and inherited risk factors. However, only the effects of comorbidities and *F5* 1601G>A did not differ between sexes, while others exhibited sexual dimorphism: overweight and *F2* 20210G>A variation were observed as significant only in females, and family history of VTE only in males. Previously, *F5* 1601G>A was identified as one of the most important non-modifiable VTE risk factors, with homozygous carriers associated with up to 20-fold higher odds of VTE development [1]. Similarly, the role of *F2* 20210G>A has been described, with the proportion of VTE cases in the population attributed to this variant being estimated as 6.2% [73]. Among non-genetic factors, overweight and positive family history have been reported to increase the overall VTE risk score by approximately 2.5-fold each [1], while comorbidities have been observed in more than 85% of all VTE patients [74]. VTE is a complex condition whose risk factors are numerous, and in this regard, our results mostly confirm what has been reported previously [1,2,3,4]. Given the lack of association between sex and VTE risk in our study, we propose that larger studies stratified in terms of age groups are needed to determine the true difference between females and males in terms of well-established VTE risk factors, as well as to confirm or oppose our findings related to *F5* 6665A>G polymorphism.

Our study has several limitations, including a relatively small sample size that decreases the statistical power of our study (especially in relation to the observed effect of rare F5 6665A>G polymorphism), as well as the lack of data on possible confounding factors that could affect the risk of VTE development, such as the use of estrogen-based oral contraceptive therapy, antiplatelet drugs, or anticoagulants. In addition, we did not take into account the types and levels of comorbidities and did not genotype the study subjects for functional polymorphisms of other relevant genes.

## 4. Materials and Methods

### 4.1. Study Participants

This retrospective observational case–control study was conducted at the Clinical Center, Podgorica, Montenegro, from 2017 to 2020. The minimum sample size was estimated based on the study by Khidri et al. [75] that investigated the association between thrombophilia-related genetic variations and preeclampsia. Employing a dominant genetic model in determining the *F5* 6665A>G genotype among 250 in Pakistani patients, the authors observed a significantly higher frequency of the A/A genotype in preeclamptic cases compared to healthy controls (87.2% vs. 76.0%, OR (95%CI): 2.17, 1.11–4.17, *p* = 0.021). Assuming a type I error rate of 0.05 and an 80% power level, and taking into account the expected frequency of the *F5* 6665A/A genotype in a European population of 92.3% [49], the sample size for our study was estimated to be 201 subjects. Our study included 103 patients (cases) diagnosed with VTE and 106 sex- and age-matched healthy controls. The inclusion criteria for the cases were the experience of at least one clinically confirmed episode of deep venous thrombosis (DVT) or pulmonary thromboembolism (PTE) and available results of diagnostic genotyping for *F2* 20210G>A (3′ UTR variant, rs1799963) and *F5* 1601G>A (Arg534Gln, rs6025) polymorphisms. Healthy controls were included if they had no history of a thromboembolic event. Neither patients nor controls could participate in the study if they were pregnant or breastfeeding, genetically related to each other, or diagnosed with cancer. Patients and controls were recruited from patients of the Clinical Centre, Podgorica, Montenegro, and from the general population, respectively, and all of them signed informed consent prior to participating in the study. The research was conducted in accordance with the Declaration of Helsinki and the Good Clinical Practice and approved by the ethics committee of the Clinical Center, Podgorica, Montenegro, decisions No 03/01-5005/1 and 03/01-13055/1.

### 4.2. Data Collection and Genotyping

Demographics and clinical characteristics, including sex, age, BMI, medical history, comorbidities, lifestyle, and habits, as well as the results of diagnostic genotyping for *F2* 20210G>A and *F5* 1601G>A, were obtained from the medical records or by a questionnaire.

DNA was extracted from whole-blood samples, using the QIA amp DNA Blood Mini Kit (Qiagen, Hamburg, Germany). Genotyping for single nucleotide polymorphisms (SNPs) of interest, namely *F2* 19911A>G (intron variant, rs3136516), *F5* 6665A>G (Asp2222Gly, rs6027; legacy numbering 6755A>G, Asp2194Gly [28]), and *F13A* 102G>T (Val34Leu, rs5985), was performed by allele-specific polymerase chain reaction (AS-PCR), using commercially available Attomol^®^ Quicktype tests (Attomol GmbH Molecular Diagnostics, Bronkow, Germany) according to the manufacturer’s instructions. The AS-PCR products were analyzed by gel electrophoresis on a 1.2% agarose gel stained with ethidium-bromide.

### 4.3. Statistical Analysis

Statistical analyses were performed using the SPSS software program, version 20 (IBM, Armonk, NY, USA). The comparison of observed with expected allele frequencies (HWE) was carried out using a chi-square test. Haplotype analysis and linkage disequilibrium testing were performed using the genetic software programs Arlequin, version 3.5 (University of Bern, Switzerland; http://cmpg.unibe.ch/software/arlequin3), and Haploview, version 4.2 (Broad Institute, Cambridge, Massachusetts, USA; http://www.broad.mit.edu/mpg/haploview/), respectively. Genotype data were presented as absolute and relative frequencies of the alleles, genotypes, and genotype groups (dominant and recessive genetic models, related to the presence of one or both minor alleles, respectively). Continuous variables were expressed as median and range. The association of investigated variables with the risk of VTE development was assessed by univariable and subsequent multivariable logistic regression analysis with backward elimination. The quality of model prediction was evaluated using the Hosmer and Lemeshow goodness-of-fit test. The odds ratios (ORs) and its 95% confidence intervals (95%CI) were calculated for independent variables to determine the strength of the observed association. To correct potential overestimation due to the relatively small sample size and to assess the robustness of the model, internal validation using a bootstrapping procedure with 1000 bootstrap resamples was performed and presented by *p*-value and 95%CI. The level of statistical significance in all the analyses was set at 0.05.

## 5. Conclusions

In conclusions, our study reports the association of *F5* 6665A>G polymorphism with increased risk of venous thromboembolism in females. In addition to all other already established VTE risk factors, we believe *F5* 6665A>G polymorphism should be considered as a possible participant in the genetic landscape of coagulation-related disorders, as well as likely a contributor to the deciphering of sex-related differences in VTE development.

## Figures and Tables

**Table 1 ijms-26-02403-t001:** Basic clinical characteristics of 209 study participants and the comparison between VTE patients (n = 103) and controls (n = 106).

Basic Characteristics	Patients	Controls	Total	OR * (95%CI)	*p*
**Sex**					
Males	49 (47.6%)	53 (50.0%)	102 (48.8%)	1.10 (0.64–1.90)	0.726
Females	54 (52.4%.)	53 (50.0%)	107 (51.2%)		
**Age**					
Median	40	43	42	0.99 (0.96–1.01)	0.221
Range	10–67	20–77	10–77		
**BMI**					
Median	26.6	25.2	26.0	1.12 (1.04–1.21)	0.003
Range	17.9–38.3	18.7–36.0	17.9–38.3		
**Overweight**					
Yes	26 (25.2%)	9 (8.5%)	35 (16.7%)	3.64 (1.61–8.22)	0.002
No	77 (74.8%)	97 (91.5%)	174 (83.3%)		
**Comorbidities**					
Yes	52 (50.5%)	6 (5.7%)	58 (27.8%)	15.12 (6.09–37.57)	<0.001
No	51 (49.5%)	100 (94.3%)	151 (72.2%)		
**Cigarette smoking**					
Ever	37 (35.9%)	41 (38.7%)	77 (36.9%)	1.05 (0.58–1.91)	0.866
Never	66 (64.1%)	65 (61.3%)	131 (62.7%)		
**Coffee consumption**					
Yes	82 (79.6%)	79 (74.5%)	161 (77.0%)	1.42 (0.74–2.73)	0.296
No	21 (20.4%)	27 (25.5%)	48 (23.0%)		
**Alcohol consumption**					
Yes	16 (15.5%)	27 (25.5%)	43 (20.6%)	1.03 (0.48–2.24)	0.932
No	87 (84.5%)	79 (74.5%)	166 (79.4%)		
**Daily physical activity**					
Yes	92 (89.3%)	101 (95.3%)	193 (92.3%)	0.41 (0.14–1.24)	0.114
No	11 (10.7%)	5 (4.7%)	16 (7.7%)		
**Family history of VTE**					
Yes	42 (40.8%)	15 (14.2%)	57 (27.3%)	4.18 (2.13–8.19)	<0.001
No	61 (59.2%)	91 (85.8%)	152 (72.7%)		

* univariate logistic regression analysis.

**Table 2 ijms-26-02403-t002:** *F2, F5*, and *F13* gene-related frequencies among study participants and the comparison between VTE patients and controls.

	Patients	Controls	Total	OR (95%CI)	*p*
***F2* 19911A>G (rs3136516)**					
**Allele**					
A	95 (46.1%)	102 (48.1%)	197 (47.1%)		
G	111 (53.9%)	110 (51.9%)	221 (52.9%)		
**Genotype**					
A/A	30 (29.1%)	22 (20.8%)	52 (24.9%)	ref.	NA
A/G	51 (49.5%)	66 (62.3%)	117 (56.0%)	0.567 (0.293; 1.097)	0.092
G/G	22 (21.4%)	18 (17.0%)	40 (19.4%)	0.896 (0.390; 2.058)	0.796
**Genotype group**					
**Dominant genetic model**					
A/A	30 (29.1)	22 (20.8)	52 (24.9)	ref.	NA
A/G + G/G	73 (70.9)	84 (79.2)	157 (75.1)	0.637 (0.338; 1.201)	0.163
**Recessive genetic model**					
A/A + A/G	81 (78.6)	88 (83.0)	169 (80.9)	ref.	NA
G/G	22 (21.4)	18 (17.0)	40 (19.1)	1.328 (0.665; 2.653)	0.422
***F5* 6665A>G (rs6027)**					
**Allele**					
A	189 (91.7)	202 (95.3%)	391(93.5%)		
G	17 (8.3%)	10 (4.7%)	27 (6.5%)		
**Genotype**					
A/A	86 (83.5%)	96 (90.6%)	182 (87.1%)	ref.	NA
A/G	17 (16.5%)	10 (9.4%)	27 (12.9%)	1.898 (0.825; 4.367)	0.132
G/G	0 (0.0%)	0 (0.0%)	0 (0.0%)	NA	NA
**Genotype group**					
**Dominant genetic model**					
A/A	86 (83.5)	96 (90.6)	182 (87.1)	ref.	NA
A/G + G/G	17 (16.5)	10 (9.4)	27 (12.9)	1.898 (0.825; 4.367)	0.132
**Recessive genetic model**					
A/A + A/G	103 (100.0)	106 (100.0)	209 (100.0)	ref.	NA
G/G	0 (0.0%)	0 (0.0%)	0 (0.0%)	NA	NA
***F13A* 102G>T (rs5985)**					
**Allele**					
G	164(79.6%)	150 (70.7%)	314 (75.1%)		
T	42 (20.4%)	62 (29.3%)	104 (24.9%)		
**Genotype**					
G/G	66 (64.1%)	55 (51.9%)	121 (57.9%)	ref.	NA
G/T	32 (31.1%)	40 (37.7%)	72 (34.4%)	0.667 (0.371; 1.199)	0.176
T/T	5 (4.9%)	11 (10.4%)	16 (7.7%)	0.379 (0.124; 1.156)	0.088
**Genotype group**					
**Dominant genetic model**					
G/G	66 (64.1%)	55 (51.9%)	121 (57.9%)	ref.	NA
G/T + T/T	37 (35.9%)	51 (48.1%)	88 (42.1%)	0.605 (0.347; 1.052)	0.075
**Recessive genetic model**					
G/G + G/T	98 (95.1%)	95 (89.6%)	193 (92.3%)	ref.	NA
T/T	5 (4.9%)	11 (10.4%)	16 (7.7%)	0.441 (0.148; 1.316)	0.142

OR—odds Ratio; 95%CI—the 95% confidence interval for the estimated OR; *p*—the probability value; ref.—reference category; NA—not applicable.

**Table 3 ijms-26-02403-t003:** Variable estimates from a multivariant regression model predicting the risk of VTE development in females (n = 107) and males (n = 102).

							Bootstrapping Analyses	
	B	S.E.	Wald χ2	*p*	OR	95%CI	*p*	95%CI
**Females**								
**Comorbidities**	5.284	1.189	19.751	<0.001	197.10	19.17; 2026.19	0.001	3.537; 37.367
**Overweight**	3.514	1.331	6.967	0.008	33.59	2.47; 456.65	0.002	0.794; 38.440
***F2* 20210G>A (rs1799963) ^a^**	3.479	1.042	11.159	0.001	32.43	4.21; 249.77	0.001	1.382; 23.692
***F5* 1601G>A (rs6025) ^a^**	4.975	1.207	16.989	<0.001	144.80	13.59; 1542.63	0.001	3.283; 35.635
***F5* 6665A>G (rs6027) ^b^**	4.160	1.264	10.824	0.001	64.06	5.38; 763.61	0.001	1.806; 24.682
Constant	−2.769	0.595	21.640	<0.001	0.06		0.001	
**Males**								
**Family history of VTE**	2.092	0.648	10.439	0.001	8.10	2.28; 28.83	0.002	0.922; 4.206
**Comorbidities**	1.843	0.613	9.047	0.003	6.32	1.90; 20.98	0.003	0.632; 3.527
***F5* 1601G>A (rs6025) ^a^**	3.001	1.098	7.466	0.006	20.10	2.34; 173.02	0.009	1.132; 22.140
Constant	−1.275	0.316	16.315	<0.001	0.28		0.001	

^a^—G/G as a reference group; ^b^—A/A as a reference group. B—the regression coefficient; S.E.—the standard error of β; Wald χ2—Wald test statistic for DF = 1; *p*—the probability value; OR—odds ratio; 95%CI—the 95% confidence interval for the estimated OR.

## Data Availability

Raw data are available at https://doi.org/10.6084/m9.figshare.27161472 (accessed on 1 February 2025).

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
