# Peer review of "F5 6665A>G Polymorphism Is Associated with Increased Risk of Venous Thromboembolism in Females"

_ijms, 2025, doi:10.3390/ijms26062403_

Round 1
Reviewer 1 Report
Comments and Suggestions for Authors
This manuscript presents novel and significant findings regarding the F5 6665A>G polymorphism and its association with VTE in females. The manuscript reports several risk factors with notably high odds ratios, accompanied by wide confidence intervals. These extreme values may stem from the small sample size of the study, which included only 103 VTE patients and 106 controls.
To address these concerns, the authors are encouraged to:
1. Provide a sample size calculation and statistical power analysis to assess whether the current sample size is sufficient to support the study's conclusions.
2. Perform sensitivity analyses for risk factors with extreme odds ratios to determine whether the findings are robust to changes in sample size or variable selection.
3. Discuss the potential impact of sample size limitations on the interpretation of results, particularly for rare polymorphisms such as F5 6665A>G.
4. If feasible, validate the findings in a larger sample or an independent cohort to enhance the reliability and generalizability of the results.
5. The introduction outlines the significance of thrombotic risk and the importance of identifying variants. Authors could cite some GWAS study identifying VTE SNP, such as PMID: 36658437, 37810217, 37076872.
Author Response
We thank the reviewer for their revision of our manuscript, and the helpful feedback they provided. We considered their suggestions in detail, and made every attempt to incorporate the amendments. Below are presented our itemized responses to specific comments:
Reviewer #1:
This manuscript presents novel and significant findings regarding the F5 6665A>G polymorphism and its association with VTE in females. The manuscript reports several risk factors with notably high odds ratios, accompanied by wide confidence intervals. These extreme values may stem from the small sample size of the study, which included only 103 VTE patients and 106 controls.
To address these concerns, the authors are encouraged to:
- Provide a sample size calculation and statistical power analysis to assess whether the current sample size is sufficient to support the study's conclusions.
Authors reply: We have now included sample size calculations in the Materials and Methods, Study participants, as follows: “The minimum sample size has been estimated based on the study by Khidri et al. (1) that investigated the association between thrombophilia-related genetic variations and preeclampsia. Employing dominant genetic model in determining F5 6665A>G genotype among 250 in Pakistani patients, the authors observed significantly higher frequency of A/A genotype in preeclamptic cases as compared to healthy controls (87.2% vs. 76.0%, OR (95% CI): 2.17, 1.11-4.17, p=0.021). Assuming type I error rate of 0.05 and 80% power level, and taking into account the expected frequency of F5 6665A/A genotype in an European population of 92.3% (2), the sample size for our study has been estimated to 201 subjects.”
- Khidri FF, Waryah YM, Ali FK, Shaikh H, Ujjan ID, Waryah AM. MTHFR and F5 genetic variations have association with preeclampsia in Pakistani patients: a case control study. BMC medical genetics. 2019;20(1):163.
- Auton A, Abecasis GR, Altshuler DM, Durbin RM, Abecasis GR, Bentley DR, et al. A global reference for human genetic variation. Nature. 2015;526(7571):68-74.
- Perform sensitivity analyses for risk factors with extreme odds ratios to determine whether the findings are robust to changes in sample size or variable selection.
Authors reply: To assess the robustness of our model, we have now performed internal validation using a bootstrapping procedure, and included the explanation and the results of the analysis in the Materials and Methods, Statistical analysis, and the Results section, Table 3, respectively. To correct potential overestimation due to the relatively small sample size, internal validation of the model using a bootstrapping procedure with 1000 bootstrap resamples was performed, and presented by p‐value and 95% CI. Resampling of data confirmed statistical significance and yielded narrower OR 95%CI, suggesting that the findings are robust to changes.
- Discuss the potential impact of sample size limitations on the interpretation of results, particularly for rare polymorphisms such as F5 6665A>G.
Authors reply: We have included the limitations of the study at the end of the Discussion section, as follows: “Our study has several limitations, including relatively small sample size that de-creases statistical power of our study (especially related to the observed effect of rare F5 6665A>G polymorphism), as well as the lack of data on possible confounding factors that could affect the risk of VTE development, such as the use of estrogen-based oral contraceptive therapy, antiplatelet drugs, or anticoagulants. In addition, we did not take into account the types and levels of comorbidities, and did not genotype study subjects for functional polymorphisms of other relevant genes.”
- If feasible, validate the findings in a larger sample or an independent cohort to enhance the reliability and generalizability of the results.
Authors reply: Since external validation is not feasible, we have performed internal validation using bootstrapping procedure with 1000 bootstrap resamples, as previously explained. The analysis indicate reliability and generalizability of the results.
- The introduction outlines the significance of thrombotic risk and the importance of identifying variants. Authors could cite some GWAS study identifying VTE SNP, such as PMID: 36658437, 37810217, 37076872..
Authors reply: The suggested publications have been cited within the Introduction section where appropriate.
The revised sections are highlighted throughout the manuscript, and a separate clean copy of the manuscript is provided. In attempt to address the valuable comments and suggestions raised, we do hope the revised version of the manuscript meets the expectations of the reviewer.
Reviewer 2 Report
Comments and Suggestions for Authors
In this manuscript, Dr. Teofilov and colleagues report the results of a study aimed at identifying the potential association between venous thrombo-embolism (VTE) and a polymorphism found in the present work in coagulation factor V (FV) gene (F5 6665A>G). I am not a bio-statistician but it seems that presence of F5 6665 G allele confers significantly higher risk of VTE among females, but rather surprisingly not among males. This F5 6665A>G missense mutation leads to the substitution of a negatively charged amino acid (i.e., Asp) with a small, neutral and highly conformationally flexible amino acid (i.e., Gly). Asp-Gly replacement at position 2222 within FV C2 domain. Position 2222 is only three amino acids from the end of the molecule (ref. 25) and close to a disulphide bridge, which can stabilize the C2 domain in the native conformation. Hence, dramatic changes of the physico-chemical properties of the amino acid at position 2222 might alter proper and functional folding of FV variant, such that secretion of active FV is reduced. Furthermore, the mutation might also affect the susceptibility of the co-factor to be proteolytically activated by thrombin or FXa or proteolytically inactivated by Protein C. The possible mechanisms underlying the effects of the F5 6665A>G mutation have not been explored in this work.
Another key issue is that the association between the F5 6665A>G mutation and VTE development was solely observed in females, but not among males. Clear-cut explanation of this effect has not been provided.
Overall, the information reported in the manuscript is important (if statistically confirmed) and may deserve publication in the Journal, even though more specialised Journals are likely more appropriate. The manuscript is well written and the Authors’ conclusions appear to be supported by the results.
Minor points
The Authors are encouraged to underline the limitations of their own work and in particular to discuss in more detail the possible mechanism(s) linking the mutation in FV to the increased risk of VTE development.
Author Response
We thank the reviewer for their revision of our manuscript, and the helpful feedback they provided. We considered their suggestions in detail, and made every attempt to incorporate the amendments. Below are presented our itemized responses to specific comments:
Reviewer #2:
In this manuscript, Dr. Teofilov and colleagues report the results of a study aimed at identifying the potential association between venous thrombo-embolism (VTE) and a polymorphism found in the present work in coagulation factor V (FV) gene (F5 6665A>G). I am not a bio-statistician but it seems that presence of F5 6665 G allele confers significantly higher risk of VTE among females, but rather surprisingly not among males. This F5 6665A>G missense mutation leads to the substitution of a negatively charged amino acid (i.e., Asp) with a small, neutral and highly conformationally flexible amino acid (i.e., Gly). Asp-Gly replacement at position 2222 within FV C2 domain. Position 2222 is only three amino acids from the end of the molecule (ref. 25) and close to a disulphide bridge, which can stabilize the C2 domain in the native conformation. Hence, dramatic changes of the physico-chemical properties of the amino acid at position 2222 might alter proper and functional folding of FV variant, such that secretion of active FV is reduced. Furthermore, the mutation might also affect the susceptibility of the co-factor to be proteolytically activated by thrombin or FXa or proteolytically inactivated by Protein C. The possible mechanisms underlying the effects of the F5 6665A>G mutation have not been explored in this work.
Authors reply: We thank the reviewer for noticing the missing link in explaining the mechanism underlying the effect of F5 6665A>G on the differentiation between two main forms of FVa. As suggested, we have provided better explanation within the Discussion section, as follows: “…closely located to accelerin glycosylation site at Asn2181…As the modulation of FVa glycosylation efficiency leads to imbalance between its two forms, we assume that the increased risk of VTE in the presence of Asp2222Gly substitution could be explained by the dominance of FVa1 over its counterpart FVa2, displaying up to 7-fold higher thrombin generation and procoagulant action (1,2). “
- Hoekema L, Castoldi E, Tans G, Girelli D, Gemmati D, Bernardi F, et al. Functional properties of factor V and factor Va encoded by the R2-gene. Thrombosis and haemostasis. 2001;85(1):75-81.
- Castoldi E, Rosing J, Girelli D, Hoekema L, Lunghi B, Mingozzi F, et al. Mutations in the R2 FV gene affect the ratio between the two FV isoforms in plasma. Thrombosis and haemostasis. 2000;83(3):362-5.
Another key issue is that the association between the F5 6665A>G mutation and VTE development was solely observed in females, but not among males. Clear-cut explanation of this effect has not been provided.
Authors reply: We do not have a clear-cut explanation, but we believe that the cause underlying this phenomenon is related to the female-biased F8 expression, which leads to higher FVIII levels in women. Therefore, the effect of F5 6665A>G variation, associated with impaired FV secretion and lower FVa-dependent activated protein C activity, which consequently lead to decreased FVIII proteolysis, is more pronounced in women as compared to men. This has been explained within the Discussion section. To further stress the importance of sexual dimorphism in disease development in general, we have also included additional paragraph, as follows: “Sexual dimorphism in disease prevalence due to sex-specific gene expression has already been reported, including the association between CFTR IVS8 poly-T variation and the se-verity of acute pancreatitis in women (1), and sex-dependent role of IFNL4 rs12979860 and rs368234815 polymorphisms in COVID-19-related pneumonia development (2). The present study is the first to report increased risk of VTE development in female carriers of F5 6665A>G variation, indicating potential predictive capacity of routine F5 genotyping in women.”
- Radosavljevic I, Stojanovic B, Spasic M, Jankovic S, Djordjevic N. CFTR IVS8 Poly-T Variation Affects Severity of Acute Pancreatitis in Women. Journal of gastrointestinal surgery : official journal of the Society for Surgery of the Alimentary Tract. 2019;23(5):975-81.
- Matic S, Milovanovic D, Mijailovic Z, Djurdjevic P, Sazdanovic P, Stefanovic S, et al. Its all about IFN-λ4: Protective role of IFNL4 polymorphism against COVID-19-related pneumonia in females. Journal of medical virology. 2023;95(10):e29152.
Overall, the information reported in the manuscript is important (if statistically confirmed) and may deserve publication in the Journal, even though more specialised Journals are likely more appropriate. The manuscript is well written and the Authors’ conclusions appear to be supported by the results.
Minor points
The Authors are encouraged to underline the limitations of their own work and in particular to discuss in more detail the possible mechanism(s) linking the mutation in FV to the increased risk of VTE development. .
Authors reply: We have included the limitations of the study at the end of the Discussion section, as follows: “Our study has several limitations, including relatively small sample size that de-creases statistical power of our study (especially related to the observed effect of rare F5 6665A>G polymorphism), as well as the lack of data on possible confounding factors that could affect the risk of VTE development, such as the use of estrogen-based oral contraceptive therapy, antiplatelet drugs, or anticoagulants. In addition, we did not take into account the types and levels of comorbidities, and did not genotype study subjects for functional polymorphisms of other relevant genes.” The possible mechanisms linking the mutation in FV to the increased risk of VTE development are included in the Discussion section, as previously mentioned.
The revised sections are highlighted throughout the manuscript, and a separate clean copy of the manuscript is provided. In attempt to address the valuable comments and suggestions raised, we do hope the revised version of the manuscript meets the expectations of the reviewer.
Reviewer 3 Report
Comments and Suggestions for Authors
L 87: All genotypes were in HWE: abbreviation mentioned in L 87 while meaning explained in L 249 (Hardy Weinberg equilibrium, HWE)
L 125 – 144: I think authors may consider moving these paragraphs to the introduction
L109: Table 3: The upper row, B stands for β – the regression coefficient?
L 147: fits well to the European frequency data provided by the 1000 Genomes Project (45). I think reference 45 refers to Ensembl (Ensembl is a registered trademarks of EMBL). However, the integrated data are available from the main website. 45 Does not appear to provide detailed information on frequency data.
Author Response
We thank the reviewer for their revision of our manuscript, and the helpful feedback they provided. We considered their suggestions in detail, and made every attempt to incorporate the amendments. Below are presented our itemized responses to specific comments:
Reviewer #3:
L 87: All genotypes were in HWE: abbreviation mentioned in L 87 while meaning explained in L 249 (Hardy Weinberg equilibrium, HWE)
Authors’ reply: Corrected.
L 125 – 144: I think authors may consider moving these paragraphs to the introduction
Authors’ reply: Although our main result is related to F5 gene polymorphism, our main goal was to assess the association between the risk of VTE development and the polymorphism of three different genes, namely F2, F5 and F13A. Therefore, in the Introduction section we intended to make all three genes equally represented, while in the Discussion, due to the differences among the observed associations, we discussed F5 in more detail. If, however, the reviewer considers it important that we move the part of the text in question from Discussion to Introduction, we will do as suggested.
L109: Table 3: The upper row, B stands for β – the regression coefficient?
Authors’ reply: Yes. It is explained in the text below the table.
L 147: fits well to the European frequency data provided by the 1000 Genomes Project (45). I think reference 45 refers to Ensembl (Ensembl is a registered trademarks of EMBL). However, the integrated data are available from the main website. 45 Does not appear to provide detailed information on frequency data.
Authors’ reply: The reference has been changed accordingly.
The revised sections are highlighted throughout the manuscript, and a separate clean copy of the manuscript is provided. In attempt to address the valuable comments and suggestions raised, we do hope the revised version of the manuscript meets the expectations of the reviewer.
Reviewer 4 Report
Comments and Suggestions for Authors
Dear authors,
I thoroughly read your manuscript ”F5 6665A>G Polymorphism is Associated with Increased Risk of Venous Thromboembolism in Females”
The abstract is well done and summarizes all the results you presented.
This manuscript is very interesting, well-designed, and organized, with detailed and organized tables explaining precisely the content of each table in the table name.
However, could you please explain why you chose three of the most frequently investigated genes in coagulation/anticoagulation pathways?
My second concern linked to this manuscript is the choice of this specific SNP among the three genes chosen. Are these polymorphisms empirically determined, or have you tried with more different polymorphisms, and these three showed acceptable results?
Could you please clearly explain the “Leiden mutation”?
This is the first study reporting a sex-dependent connection between F5 6665A>G genetic polymorphism (female) and VTE development. Overweight and F2 20210G>A variation was observed as significant only in females. There is a possible explanation for F5 polymorphism, but no information exists for F2 20210G>A and comorbidities variation.
FV 666A>G mutation linked to exaggerated venous thromboembolism which is observed in this study was sex-dependent, as the authors were able to identify it only among females. Could you please emphasize the explanation of this novelty?
F5 6665A>G is a missense variant leading to amino acid Asp>Gly replacement. What is your opinion on why the two main activated forms of FVa make FVa1 more thrombogenic than FVa2?
This is a very interesting manuscript with an interesting theme, but I have to repeat the obvious limitation. There is only one interesting novelty explaining the possible FV 666A>G mutation observed among females linked to an increased risk of VTE. The proposed explanation could clarify the observed association between F5 6665A>G linked to increased risk of VTE. Could you propose the presumed mechanism, in detail including sex-specific aspects in the context of FVIII, female-biased F8 gene expression? You gave us an explanation in the discussion, but could you nevertheless insist on the novelty, because that makes this manuscript acceptable? The sexual dimorphism is a very important result, so you should emphasize that.
Best regards
Author Response
We thank the reviewer for their revision of our manuscript, and the helpful feedback they provided. We considered their suggestions in detail, and made every attempt to incorporate the amendments. Below are presented our itemized responses to specific comments:
Reviewer #4:
Dear authors, I thoroughly read your manuscript ”F5 6665A>G Polymorphism is Associated with Increased Risk of Venous Thromboembolism in Females” The abstract is well done and summarizes all the results you presented. This manuscript is very interesting, well-designed, and organized, with detailed and organized tables explaining precisely the content of each table in the table name. However, could you please explain why you chose three of the most frequently investigated genes in coagulation/anticoagulation pathways?
Authors’ reply: Most of the previous investigations linked VTE risk with the polymorphism of genes associated with coagulation/anticoagulation pathways, as already mentioned in the Introduction section. In our study, these particular genes were chosen based on reported functionality of their polymorphisms in terms of other coagulation-dependent diseases, such as ischemic stroke, myocardial infarction, or peripheral arterial occlusive disease (1). This reference is now included in the Introduction section.
- Herm J, Hoppe B, Siegerink B, Nolte CH, Koscielny J, Haeusler KG. A Prothrombotic Score Based on Genetic Polymorphisms of the Hemostatic System Differs in Patients with Ischemic Stroke, Myocardial Infarction, or Peripheral Arterial Occlusive Disease. Frontiers in cardiovascular medicine. 2017;4:39.
My second concern linked to this manuscript is the choice of this specific SNP among the three genes chosen. Are these polymorphisms empirically determined, or have you tried with more different polymorphisms, and these three showed acceptable results?
Authors’ reply: As explained in the Introduction section, we have chosen polymorphisms that have been reported to be both frequent in the population, and either proven functional (“…frequent and functional intronic variant 19911A>G…’’), or whose function is expected to be important, but still not confirmed (“one of the most intriguing seems to be 6665А>G, located within the functional domain of the gene…and whose role in VTE development is still unresolved”; “…102G>T, located near thrombin activation site, has been reported as protective against venous thrombosis, but contradictory findings have been published as well…”). We have reported all the results obtained in the study, regardless if they were found significant or not.
Could you please clearly explain the “Leiden mutation”?
Authors’ reply: This variant causes reduced factor V degradation, causing hypercoagulability, and consequently increasing the risk of thromboembolic disease. This is now included in the Introduction section, followed by the appropriate references.
This is the first study reporting a sex-dependent connection between F5 6665A>G genetic polymorphism (female) and VTE development. Overweight and F2 20210G>A variation was observed as significant only in females. There is a possible explanation for F5 polymorphism, but no information exists for F2 20210G>A and comorbidities variation.
Authors’ reply: We have now included an explanation on the association of F2 20210G>A, F5 1601G>A, comorbidities, overweight, and family history of VTE with the increased risk of VTE development, as follows: “Previously, F5 1601G>A has been identified as one of the most important non-modifiable VTE risk factors, with homozygous carriers associated to up to 20-fold higher odds of VTE development (1). Similarly, the role of F2 20210G>A have been described, with the proportion of VTE cases in the population attributed to this variant being estimated to 6.2% (2). Among non-genetic factors, overweight and positive family history have been reported to increase the overall VTE risk score by approximately 2.5-fold each (1), while comorbidities have been observed in more than 85% of all VTE patients (3). “
- Crous-Bou M, Harrington LB, Kabrhel C. Environmental and Genetic Risk Factors Associated with Venous Thromboembolism. Seminars in thrombosis and hemostasis. 2016;42(8):808-20.
- Gohil R, Peck G, Sharma P. The genetics of venous thromboembolism. A meta-analysis involving approximately 120,000 cases and 180,000 controls. Thrombosis and haemostasis. 2009;102(2):360-70.
- Kroep S, Chuang LH, Cohen A, Gumbs P, van Hout B, Monreal M, et al. The impact of co-morbidity on the disease burden of VTE. Journal of thrombosis and thrombolysis. 2018;46(4):507-15.
FV 666A>G mutation linked to exaggerated venous thromboembolism which is observed in this study was sex-dependent, as the authors were able to identify it only among females. Could you please emphasize the explanation of this novelty?
Authors’ reply: We have included additional explanation in regard to the novelty of our findings as follows: “Sexual dimorphism in disease prevalence due to sex-specific gene expression has already been reported, including the association between CFTR IVS8 poly-T variation and the se-verity of acute pancreatitis in women (1), and sex-dependent role of IFNL4 rs12979860 and rs368234815 polymorphisms in COVID-19-related pneumonia development (2). The present study is the first to report increased risk of VTE development in female carriers of F5 6665A>G variation, indicating potential predictive capacity of routine F5 genotyping in women.”
- Radosavljevic I, Stojanovic B, Spasic M, Jankovic S, Djordjevic N. CFTR IVS8 Poly-T Variation Affects Severity of Acute Pancreatitis in Women. Journal of gastrointestinal surgery : official journal of the Society for Surgery of the Alimentary Tract. 2019;23(5):975-81.
- Matic S, Milovanovic D, Mijailovic Z, Djurdjevic P, Sazdanovic P, Stefanovic S, et al. Its all about IFN-λ4: Protective role of IFNL4 polymorphism against COVID-19-related pneumonia in females. Journal of medical virology. 2023;95(10):e29152.
F5 6665A>G is a missense variant leading to amino acid Asp>Gly replacement. What is your opinion on why the two main activated forms of FVa make FVa1 more thrombogenic than FVa2?
Authors’ reply: As explained (now in more detail) within the Discussion section, impaired FV secretion associated with the presence of Asp2222Gly substitution could be explained by protein misfolding, which results in its retention and subsequent degradation within the endoplasmic reticulum (1). In addition, this variation is also closely located to accelerin glycosylation site (2), which is responsible for differentiation between two main forms of FVa, namely FVa1 and FVa2: the former is more glycosylated, thus associated with impaired membrane binding and lower APC-cofactor activity (2,3). As the modulation of FVa glycosylation efficiency leads to imbalance between its two forms, we assume that the increased risk of VTE in the presence of Asp2222Gly substitution could be explained by the dominance of FVa1 over its counterpart FVa2, displaying up to 7-fold higher thrombin generation and procoagulant action (4,5).
- Yamazaki T, Nicolaes GAF, Sørensen KW, Dahlbäck B. Molecular basis of quantitative factor V deficiency associated with factor V R2 haplotype. Blood. 2002;100(7):2515-21.
- Nicolaes GA, Dahlbäck B. Factor V and thrombotic disease: description of a janus-faced protein. Arteriosclerosis, thrombosis, and vascular biology. 2002;22(4):530-8.
- Schreuder M, Reitsma PH, Bos MHA. Blood coagulation factor Va's key interactive residues and regions for prothrombinase assembly and prothrombin binding. Journal of thrombosis and haemostasis : JTH. 2019;17(8):1229-39.
- Hoekema L, Castoldi E, Tans G, Girelli D, Gemmati D, Bernardi F, et al. Functional properties of factor V and factor Va encoded by the R2-gene. Thrombosis and haemostasis. 2001;85(1):75-81.
- Castoldi E, Rosing J, Girelli D, Hoekema L, Lunghi B, Mingozzi F, et al. Mutations in the R2 FV gene affect the ratio between the two FV isoforms in plasma. Thrombosis and haemostasis. 2000;83(3):362-5.
This is a very interesting manuscript with an interesting theme, but I have to repeat the obvious limitation. There is only one interesting novelty explaining the possible FV 666A>G mutation observed among females linked to an increased risk of VTE. The proposed explanation could clarify the observed association between F5 6665A>G linked to increased risk of VTE. Could you propose the presumed mechanism, in detail including sex-specific aspects in the context of FVIII, female-biased F8 gene expression? You gave us an explanation in the discussion, but could you nevertheless insist on the novelty, because that makes this manuscript acceptable? The sexual dimorphism is a very important result, so you should emphasize that.
Authors’ reply: We have now emphasized the sexual dimorphism at the end of the Discussion section, as suggested.
The revised sections are highlighted throughout the manuscript, and a separate clean copy of the manuscript is provided. In attempt to address the valuable comments and suggestions raised, we do hope the revised version of the manuscript meets the expectations of the reviewer.
Round 2
Reviewer 4 Report
Comments and Suggestions for Authors
Dear Authors,
The manuscript is significantly improved. You explained clearly why you used those genes and polymorphisms. What is most important is that the authors emphasized sexual dimorphism as the novelty of this manuscript.
Best regards